# Rational Fabrication of Defect-Rich and Hierarchically Porous Fe-N-C Nanosheets as Highly Efficient Oxygen Reduction Electrocatalysts for Zinc-Air Battery

**DOI:** 10.3390/molecules28072879

**Published:** 2023-03-23

**Authors:** Sensen Li, Yan Lv, Sawida Elam, Xiuli Zhang, Zhuojun Yang, Xueyan Wu, Jixi Guo

**Affiliations:** State Key Laboratory of Chemistry and Utilization of Carbon Based Energy Resources, College of Chemistry, Xinjiang University, Urumqi 830017, China

**Keywords:** electrocatalytic, Fe-N-C materials, oxygen reduction reaction, two-dimensional materials

## Abstract

The rational design of morphology and structure for oxygen reduction reaction (ORR) catalysts still remains a critical challenge. Herein, we successfully construct defect-rich and hierarchically porous Fe-N-C nanosheets (Fe-N-CNSs), by taking advantage of metal-organic complexation and a mesoporous template. Benefiting from the advantages of high density of active sites, fast mass transfer channels, and sufficient reaction area, the optimal Fe-N-CNSs demonstrate satisfactory ORR activity with an excellent half-wave potential of up to 0.87 V, desirable durability, and robust methanol tolerance. Noteworthy, the Fe-N-CNSs based zinc–air battery shows significant performance with a peak power density of 128.20 mW cm^−2^ and open circuit voltage of 1.53 V, which reveals that the Fe-N-CNSs catalysts present promising practical application prospects. Therefore, we believe that this research will provide guidance for the optimization of Fe-N-C materials.

## 1. Introduction

Increasing demand for energy and the massive consumption of non-renewable energy sources have prompted us to explore new types of energy storage devices [1,2,3]. With excellent theoretical energy density (1086 Wh kg^−1^), affordability, and environmental friendliness, zinc–air batteries (ZABs) have been considered as among the most potential candidates to cope with the energy requirements. The electrochemical performances of ZABs are primarily dependent on the efficiency of oxygen reduction reaction (ORR) [4,5,6,7]. Nevertheless, the sluggish kinetic of ORR suffering from a multiple electron transfer causes a confined output power and energy storage efficiency of ZABs. It is, therefore, essential to create an effective ORR catalyst to promote its slow kinetics, reduce the overpotential, and improve energy storage efficiency the of ZABs. The most effective ORR catalysts have been traditionally regarded as Pt-based ones due to a low binding energy with oxygen [8,9]. The prohibitive cost and inferior stability, however, prevent its large-scale application in air cathodes [10]. Therefore, designing and exploring cost-effective, efficient, and durable ORR catalysts is highly vital. Metal-nitrogen-carbon (M-N-C, M = Fe, Co, Ni, Cu, etc.) catalysts were already extensively studied as ORR catalysts [11,12,13,14]. The transition metals with unfilled d orbitals can bind better to oxygen intermediates and thus exhibit excellent ORR native positive activity [15,16,17,18]. In particular, Fe-N-C with the encouraging ORR activity and excellent catalytic durability has been deemed as the most prospective replacement for Pt-based electrocatalysts [19,20,21,22,23]. Among all 3d-transition metal elements, Fe shows the closest-to-optimum OH* binding energy. After O_2_ is adsorbed to the centre of the Fe atom, the electrons in its 2π orbital are transferred to the hole d_z_^2^ orbital of Fe and the partially filled d_xy_ or d_xz_ is back-bonded to the 2π* orbital of oxygen [24,25,26,27,28,29,30]. However, the micromorphology and nanostructure of numerous reported Fe-N-C materials are inhomogeneous, which leads the catalytic site to be challenging to expose to reduction of catalytic activity [31,32,33]. To address this issue, the reasonable design of the micromorphology and nanostructure of Fe-N-C materials is fundamental for constructing highly efficient Fe-N-C catalysts [23,34,35,36,37].

Two-dimensional (2D) materials were intensively explored as ORR catalysts because of large specific surface area, unique physicochemical properties as well as thin thickness, which can provide rapid mass transfer channels, an abundance of active sites, and sufficient reaction area [38,39,40,41]. Typically, 2D materials are synthesized by top-down or bottom-up methods, including chemical vapor deposition (CVD), physical/chemical exfoliation, and wet chemical synthesis [42,43,44,45]. The simple and green template strategy has been widely applied in the fabrication of 2D materials because of its ability to construct specific morphologies and hierarchical porous structures [46,47]. In addition, the construction of 2D materials with a defect-rich and hierarchical porous structures is highly desirable and challenging for improving ORR performance [48,49]. For instance, the group of Wang prepared Fe-N-C nanosheets using layered montmorillonite (MMT) as a template with a half-wave potential (E_1/2_) of 0.87 V [50]. The corresponding ZAB, assembled with the obtained material as a catalyst, exhibited a peak power density of 92.5 mW cm^−2^. Nevertheless, MMT cannot provide abundant micro/mesopores or defects to further enhance ORR performance.

Inspired by the above analysis, herein, we successfully synthesized Fe-N-C nanosheets (Fe-N-CNSs), by employing metal-organic complexation and a mesoporous magnesium carbonate base (MCB) template. The obtained Fe-N-CNSs catalysts featured a defect-rich, hierarchically porous, and high specific surface area (1303.43 m^2^ g^−1^), which can provide an abundance of active sites, mass transfer channels, as well as sufficient reaction area. The Fe-N-CNSs catalysts exhibited excellent ORR performance (E_1/2_ = 0.87 V), surpassing commercial 20% Pt/C (E_1/2_ = 0.85 V). In addition, the ZABs were assembled with Fe-N-CNSs as the catalysts for the air cathode, which demonstrated superior performance compared to Pt/C, reflecting the promising potential of Fe-N-CNSs in practical applications. The strategy that was developed in this work provides a universal approach to optimize Fe-N-C catalysts.

## 2. Results and Discussion

Figure 1 depicts the construction path of Fe-N-CNSs. Owing to the cheating ability, ethylene diamine tetra acetic acid (EDTA) can form complexes with Fe^3+^. Basic magnesium carbonate (MCB) is decomposed into MgO, CO_2_, and H_2_O in a high temperature pyrolysis process. The generated MgO is used as a mesoporous template that can induce the polymerization of small molecules on its surface, while CO_2_ and H_2_O escaping from the carbon substrate can create abundant micropores and produce exfoliation, which promotes the construction of sheet-like structures [51,52]. The scanning electron microscopy (SEM) analysis was performed on the samples that were acquired to examine their micromorphology. The pyrolysis temperature can significantly affect the structure and micromorphology of carbon-based products. Hence, we investigate how the pyrolysis temperatures affect the micromorphology of the materials. When the pyrolysis temperature is 800 °C, Fe-N-CNSs-800 is mainly composed of carbon fragments and carbon blocks, showing an irregular curled sheet morphology, as shown in Appendix A, which might be due to insufficient activation. As a result, the substance becomes less porous and has fewer mass transfer pathways. Noteworthy, as shown in Figure 1a and Appendix A, Fe-N-CNSs-900 shows a cross-linked network consisting of 2D porous nanosheets. The large surface area of materials provided by the 2D nanosheet structure makes it simpler to reveal active sites, while the hierarchically porous structure offers a high density of active sites and a significant number of rapid mass transfer pathways. As a result, we anticipate that the produced 2D hierarchically porous Fe-N-C nanosheets will function satisfactorily in ORR. Nevertheless, for Fe-N-CNSs-1000, as shown in Appendix A, the disruption of the sheet-like structures results in fewer catalytic sites. According to the aforementioned findings, one of the crucial elements in the production of a hierarchically porous nanosheet structure is a proper pyrolysis temperature. With regard to Fe-N-CNSs, a hierarchically porous nanosheet structure is best when the pyrolysis temperature is 900 °C. In addition, we look at the morphology of a sample of Fe-N-C-900 to reveal how MCB affects material morphology. It presents a distinctly larger bulk structure, as shown in Appendix A, without lamellar structures or pores being found. The above results indicate that the introduction of MCB can generate 2D porous nanosheet structures, which reflects the dual efficacy of MCB as a template and an activator.

Subsequently, the microscopic morphology of Fe-N-CNSs was explored by transmission electron microscopy (TEM). The Fe-N-CNSs shows a thin sheet-like morphology and the presence of many mesopores on the surface, as shown in Figure 1b and Appendix A, without Fe-associated nanoparticles or nanoclusters, which provides a high surface area and fast mass transfer channels. Figure 1c illustrates the existence of lattice distortion defects in graphitic carbon in the high-resolution transmission electron microscopy (HRTEM) image of Fe-N-CNS, which proves that the heteroatoms are doped into the carbon substrate [53]. In addition, lattice stripes with a lattice spacing of 0.34 nm, which corresponds to the (002) crystal plane of graphitic carbon, are found in the HRTEM images. Further analysis of Fe-N-CNSs was carried out using HADDF-STEM and EDS mapping in conjunction with TEM. As shown in Figure 1d, the high angle angular dark field-scanning transmission electron microscopy (HADDF-STEM) picture clearly shows the existence of pores and hollow structures, which serves as further proof that porous structures were successfully prepared. The results associated element mapping and confirmed that Fe, N, and C elements are all uniformly dispersed in Fe-N-CNSs, proving that Fe and N have been effectively doped into the carbon skeleton. Additionally, the creation of Fe-N_x_ sites is supported by the substantial overlap between the dispersion of the Fe and N elements.

X-ray diffraction (XRD) was used to evaluate the catalysts in order to better identify their crystalline structure. As shown in Figure 2a, several catalysts exhibit characteristic peaks of the (002) and (101) planes of carbon at 23° and 44°, which corresponds to the classical carbon structure [54]. Meanwhile, the absence of characteristic peaks of Fe and its compounds further confirms the inexistence of highly crystalline Fe-containing species in Fe-N-CNSs catalysts. As a result, we can assume that Fe-N_x_ may make up the majority of the Fe element in Fe-N-CNSs. By using X-ray photoelectron spectroscopy (XPS), the chemical makeup of the catalyst surface was examined. As shown in Figure 2b, XPS surveys show obvious signals of C, N, and O elements; nevertheless, the signal of the Fe element is undetected, which might be attributed to the low content of elemental Fe, and the findings are aggregated in Table 1. As shown in Appendix A, the C 1s XPS spectra are deconvoluted into five peaks, which are indexed to C-C (284.5 eV, sp^2^ hybridized carbon), C-C (285.2 eV sp^3^ disordered carbon), C-N (286.0 eV), C-O (286.5 eV), and C=O (287.1 eV), accordingly [55]. The presence of the C-N peak demonstrates that the N atom has been successfully doped into the carbon skeleton. In addition, the ratio of C-C (sp^2^) and C-C (sp^3^) (sp^2^/sp^3^) reflects the degree of defects in the carbon-based material. The Fe-N-CNSs shows a sp^2^/sp^3^ of 0.99, lower than the N-CNSs (1.11) and Fe-N-C (1.08), indicating that Fe-N-CNSs catalyst is rich in defects. The Fe 2p XPS spectrum of Fe-N-CNSs presents two groups of peaks for Fe^2+^ (711.3 and 724.5 eV) and Fe^3+^ (716.0 and 729.1 eV), and no obvious Fe^0^ appeared, as shown in Appendix A, further proving that no Fe or its compounds are formed. As shown in Figure 2c, to further determine the existing configuration of the N element, the N 1s spectra are deconvoluted into five peaks and the findings are aggregated in Table 2. The N 1s spectra of the catalysts exhibit five peaks at about 398.3, 399.2, 400.3, 401.3 and 402.7 eV that, respectively, correspond to pyridinic N, Fe-N, pyrrolic N, graphitic N, and oxidized N [56]. As there is a Fe-N peak, Fe-N_x_ sites have probably been formed, and Fe-N_x_ sites are the most dominant sites for Fe-N-C catalysts [57]. Additionally, the pyridine N content in Fe-N-CNSs reaches up to 41.31%. In addition to the electrons provided to the conjugated π-bond, pyridine N also contains a pair of lone electrons that facilitate the adsorption of O_2_, thereby increasing the onset potential [58,59]. The inductively coupled plasma optical emission spectrometry (ICP-OES) results show that Fe-N-CNSs presents a higher Fe content of 0.24% compared to Fe-N-C (0.07%).

Subsequently, the extent of defects in the carbon structure of the catalyst was investigated by Raman spectroscopy. As shown in Figure 2d and Appendix A, the D and G bands at 1341 cm^−1^ and 1590 cm^−1^, which stand for lattice defects and carbon atom graphitization, accordingly, are clearly visible in the Raman spectra. For the Raman spectrum, the ratio of the D peak and G peak intensity (I_D_/I_G_) can reflect some extent the level of disorder or defects in the structure of carbon-based materials [60]. The Fe-N-CNSs exhibits an I_D_/I_G_ of 1.04, higher than the N-CNSs (0.96) and Fe-N-C (0.98), indicating that the introduction of a templating agent increases the defect degree of the material, and these abundant defect sites may create more catalytic sites, thereby enhancing the catalytic efficiency.

Via a series of SEM and TEM characterizations, the macro- and mesoporous structure of materials can be directly viewed; nonetheless, the smaller pore size micro-pores cannot be seen. Therefore, the pore structure of the obtained materials was further characterized by N_2_ adsorption and desorption measurements with the outcomes noted in Table 3. All catalysts present an IV isotherm with a distinct hysteresis loop, which confirm the existence of abundant mesopores, as shown in Figure 2e and Appendix A. The Fe-N-CNSs-900 presents the highest surface area (1303.43 m^2^ g^−1^) among all the constructed materials, and this value significantly outperforms Fe-N-C (421.15 m^2^ g^−1^). The Fe-N-CNSs displays such a large surface area due to its rich porosity and distinctive 2D nanosheet structure. As shown in Figure 2f and Appendix A, from the pore size distribution curve, the Fe-N-CNSs demonstrates more abundant mesopores and micropores than Fe-N-C. These results indicate that MCB as a template agent and activation agent can produce a unique 2D nanosheet morphology, abundance of micro/mesopores, and ultrahigh specific surface area. The numerous micropores deliver more abundant active sites, while mesopores facilitate the transport of reaction intermediates. The high specific surface area provides sufficient reaction area and exposes more active surface sites.

The synthesized Fe-N-CNSs materials are expected to be excellent ORR catalysts due to a defect-rich, hierarchical porous, and ultrahigh surface area. Using cyclic voltammetry (CV), the Fe-N-CNSs is first investigated. As shown in Figure 3a, an obvious oxygen reduction peak is observed at approximately 0.8 V (vs. RHE) in the O_2_ saturated electrolyte, while it is not observed in the saturated N_2_ electrolyte, demonstrating that Fe-N-CNSs exhibits positive activity toward ORR in O_2_ saturated electrolytes. The ORR activity of these materials was further characterized by linear scanning voltammetry (LSV). As shown in Figure 3b,c, the target catalyst, Fe-N-CNSs, exhibits the highest ORR activity, including an excellent E_1/2_ (0.87 V) and kinetic current density at 0.8 V (j_k_@0.8 V = 27.66 mA cm^−2^). The ORR activity of Fe-N-CNSs is not only better than that of the Fe-N-C catalyst but better than that of the Pt/C catalyst (E_1/2_ = 0.85 V, j_k_@0.8 V = 20.61 mA cm^−2^) and many reported Fe-N-C catalysts (Appendix A). The carbonation temperature can affect the catalytic performance of carbon-based materials to some extent, so we perform a comparative analysis for the ORR performance of the samples at different temperatures. The E_1/2_ of the three catalysts reach 0.83, 0.87 and 0.84 V, respectively, and the Fe-N-CNSs-900 catalysts present the highest E_1/2_, as shown in Appendix A. Additionally, the amount of Fe (NO_3_)_3_·9H_2_O added is investigated to obtain better experimental conditions, as shown in Appendix A, and the highest ORR performance is observed for Fe-N-CNSs-30. The LSV curves are recorded at different rotational speeds, as shown in Figure 3d. The Koutecky–Levich (K–L) plots present several fitted lines with almost identical slopes. The half-wave potential steadily declines while the limiting current density gradually rises as the speed grows, representing the kinetics of vortex diffusion during ORR. Depending on the slope, in the range of 0.3, 0.4, 0.5 and 0.6 V, the average number of electrons transferred (n) for Fe-N-CNSs are calculated to be about 3.94, which reveals that Fe-N-CNSs obeys the first-order kinetics and approximately a 4e^-^ pathway. Electrochemical impedance spectroscopy (EIS) was performed to further analyze the kinetics of the ORR process in the material. As shown in Appendix A, Fe-N-CNSs exhibits smaller semicircle diameters than Pt/C in the low frequency range, which indicates that Fe-N-CNSs has faster transfer kinetics at the electrode/electrolyte interface due to its high density of active sites. Furthermore, in the high frequency range, the Fe-N-CNSs has steeper tilt lines, demonstrating that Fe-N-CNSs has a better diffusion rate at the reaction interface due to its hierarchically porous nanosheet structure.

An excellent ORR catalyst requires not only excellent ORR activity but robust methanol tolerance and remarkable stability. The durability and methanol resistance of Fe-N-CNSs catalysts are examined by chronoamperometric i-t measurements. As a comparison, we also test commercial 20% Pt/C using the same method. As shown in Figure 3e, after 400 s of 3 mL methanol addition to the electrolyte, the relative current of the commercial 20% Pt/C catalysts show a significant drop, while the Fe-N-CNSs catalysts remain almost unchanged. The result indicates the Fe-N-CNSs exhibits a strong methanol tolerance. Moreover, the Fe-N-CNSs catalysts still show 90% current retention after 12,000 s of cycling, as shown in Figure 3f, indicating that the Fe-N-CNSs catalysts are highly stable. The catalysts display exceptional ORR performance, owing to the advantages of abundant catalytic sites, sufficient reaction area and fast mass transfer channels. To investigate the catalytic sites of the Fe-N-CNSs, KSCN is added to poison the Fe-N_x_ sites during ORR as SCN^-^ has a strong affinity for Fe^3+^. As shown in Appendix A, the ORR performance drops obviously after adding of KSCN, indicating that Fe-N_x_ is the most critical catalytic site for Fe-N-CNSs. Notably, the half-wave potential still remains at a high level after the addition of KSCN, indicating that the C-N sites also contribute to increased catalytic activity. Combined with the analysis of the XPS data, the following conclusions are drawn: Fe-N_x_ and pyridinic N contribute more to ORR, and the higher their content, the better the ORR activity.

Considering the splendid ORR performance, we assemble a Fe-N-CNSs-based zinc–air battery (ZAB), as shown in Figure 4a, to further explore its practical application prospects. As shown in Figure 4b, the open circuit voltage (OCV) of the assembled ZAB is tested separately by the open circuit potential time measurement (OCPT) and an electronic multimeter. The Fe-N-CNSs-based ZAB can sustain a high OCV of 1.53 V, which outperforms commercial 20% Pt/C-based ZABs (1.44 V). Using the assembled Fe-N-CNSs-based ZAB as a power source, an LED bulb (1.2 V) with the word “XJU” can be lit, as shown in Figure 4c, which further confirms that the Fe-N-CNSs materials present an excellent prospect of practical application. The Fe-N-CNSs-based ZAB exhibits a peak power density of 128.20 mW cm^−2^, as shown in Figure 4d, which exceeds the Pt/C-based ZAB (101.80 mW cm^−2^) and outperforms most of the reported ZABs assembled from Fe-N-C materials (Appendix A). The discharge voltage of Fe-N-CNSs-based ZAB remains stable at all current densities (5–50 mA cm^−2^) and still maintains 1.10 V, as shown in Figure 4e, which outperforms the Pt/C-based ZAB. The above results reveal that Fe-N-CNSs-based ZAB exhibits a better capacity and multiplicative performance. Furthermore, the specific capacity of the Fe-N-CNSs-based ZAB reaches a high energy density of 746.37 mA h g_Zn_^−1^, which compares favorably with the Pt/C-based ZAB (713.35 mA h g_Zn_^−1^), as shown in Figure 4f.

## 3. Materials and Methods

### 3.1. Materials

Magnesium carbonate basic (MCB), ethanol, Fe (NO_3_)_3_·9H_2_O, ethylene diamine tetra acetic acid (EDTA), commercial 20% Pt/C, concentrated hydrochloric acid (HCl), 0.5 wt% Nafion, concentrated hydrochloric acid (HCl) and methanol were all purchased from Alfa Aesar and Aladdin chemical reagent company. All of the compounds were utilized directly after delivery without additional purification and are of analytical grade.

### 3.2. Catalyst Preparation

First, 0.5 g of MCB was added to 40 mL of absolute ethanol and ultrasonically dispersed for 0.5 h. Then, 0.8 g EDTA and 30 mg Fe (NO_3_)_3_·9H_2_O were added and stirring was continued for 5 h. Subsequently, the above mixture was dried in an oven at 60 °C to remove ethanol, and the resulting pale–yellow solid was ground into a homogeneous powder. Finally, the precursor was pyrolyzed in a tube furnace under N_2_ atmosphere at 900 °C for 2 h at a heating rate of 5 °C min^−1^. The resulting black solid was added to 60 mL of 2 M HCl solution and stirred for 12 h. The prepared material was washed three times by centrifugation with deionized water and absolute ethanol, respectively, and then dried in a vacuum oven overnight. The prepared samples were labeled as Fe-N-CNSs (also called Fe-N-CNSs-900 and Fe-N-CNSs-30). For comparison, the precursors were also pyrolyzed at 800 °C and 1000 °C, and the prepared carbon materials were labeled as Fe-N-CNSs-x (x = 800 and 1000). The addition of the Fe source was adjusted, and the resulting materials were labelled Fe-N-CNSs-m (m = 15, 30, 60). In addition, the precursors without Fe (NO_3_)_3_·9H_2_O or MCB were pyrolyzed at 900 °C, and the prepared materials were labeled as N-CNSs or Fe-N-C.

### 3.3. Physical Characterizations

The micromorphology of the materials was observed by SEM (SU-4800, Hitachi, Japan) and TEM (JEM-2100F, JEOL, Japan). The crystal structure of the materials was analyzed by XRD (Bruker D8, using filtered Cu Kα radiation). The degree of defects in the materials was analyzed by a Raman spectrometer (Vertrex 70, Bruker, Germany) with a laser excitation wavelength of 532 nm. The exact content of elemental Fe in the material was obtained by ICP-OES (Optima 8000, PerkinElmer, USA). N2 adsorption/desorption tests were conducted at 77 K by a gas adsorption analyzer (ASAP 2020, Micromeritics, USA). The specific surface area and pore size of the samples were calculated by Brunauere–Emmette–Teller (BET) and Barrett–Joyner–Halenda (BJH) models, respectively. The chemical composition of the catalyst was analyzed by XPS (ESCALAB 250, Thermo Scientific, USA).

### 3.4. Electrochemical Measurements

In a standard three-electrode setup, electrochemical experiments were carried out at room temperature. The electrochemical workstation is CHI 760E. The 0.1 M KOH solution saturated with O_2_/N_2_ serves as the electrolyte. The working electrode, counter electrode, and reference electrode that we employed were a spinning disk ring electrode, a platinum wire electrode, and a saturated calomel electrode, respectively. The working electrode was polished with Al_2_O_3_ polishing powders with diameters of 1 μm and 50 nm, respectively. To create a well-dispersed ink, 2.5 mg of catalyst and 20 μL of 0.5 wt% Nafion were dispersed in 480 μL of ethanol. Sonication was then applied for 30 min. A loading of 254.78 μg cm^−2^ was achieved by adding 10 μL of catalyst ink dropwise to the working electrode and drying it at room temperature. Comparatively, the working electrode was loaded with commercial 20% Pt/C, which was applied dropwise. The potentials mentioned in this work are relative to the reversible hydrogen electrode (RHE). Using a scan rate of 50 mV s^−1^, the CV curves of catalysts were examined in N_2_ or O_2_ saturated 0.1 M KOH solution. In an O_2_ saturated 0.1 M KOH solution, the LSV curves of the catalysts were examined at different rotational speeds of 400, 625, 900, 1225, 1600 and 2025 rpm with a scan rate of 10 mV s^−1^. The number of transferred electrons (*n*) can be calculated by the Koutecky–Levich (K–L) formula:$$\frac{1}{j}=\frac{1}{j_{k}}+\frac{1}{j_{l}}=\frac{1}{j_{k}}+\frac{1}{{B\omega }^{1/2}}$$
$$B=0.2nFC_{0}D_{0}^{2/3}\upsilon^{-1/6}$$

In the formula, *j* is the measured current density, *j_l_* is the limiting current density, *j_k_* is the kinetic current density, and the calculation formula is *j_k_* = (*j_l_
*× *j*)/(*j_l_* − *j*). *ω* is the rotation rate of the rotating disk electrode (RDE). *B* is determined by the slope of the K–L equation, *n* is the number of transferred electrons, *F* is the Faraday constant (96,485 C mol^−1^), *C*_0_ is the volume concentration of O_2_ (1.2 × 10^−6^ mol cm^−3^), *D*_0_ is the O_2_ at 0.1 M of the diffusion coefficient in KOH (1.9 × 10^−5^ cm^2^ s^−1^), *υ* is the dynamic viscosity (0.01 cm^2^ s^−1^). A constant of 0.2 is adopted when the rotation rate is expressed in rpm.

Electrochemical impedance spectroscopy (EIS) was performed in 0.1 M KOH by applying an AC voltage with 5 mV amplitude in frequencies ranging from 0.01 Hz to 100 kHz. The specific data can be fitted by Zview software.

### 3.5. Zn-Air Battery Test

The performance characterization of all ZABs were performed on an electrochemical workstation (CHI 760E). The zinc–air batteries used 6 M KOH as the electrolyte, zinc foil as the negative electrode and carbon fiber paper (3 mg cm^−2^) coated with catalyst ink as the positive electrode. Following this, 3 mg of Fe-N-CNSs and 0.3 mg of acetylene black were dispersed in 480 μL of ethanol and 20 μL of 5% Nafion solution, followed by sonication for 30 min to obtain a well-dispersed ink. For comparison, a Pt/C-based air cathode was prepared using the same method.

## 4. Conclusions

In summary, we successfully fabricated Fe-N-C nanosheets (Fe-N-CNSs) by a simple, versatile, and green template synthesis strategy. The optimal Fe-N-CNSs catalysts present the large surface area, hierarchical porous and defect-rich structure, which provides sufficient reaction area, mass transfer channels, as well as abundance of active sites, accelerating kinetic ORR activity on Fe-N-CNSs, along with a significant E_1/2_ of up to 0.87 V and strong stability (superior to Pt/C). Meanwhile, the Fe-N-CNSs-based ZAB also exhibits superior performance with a peak power density of 128.20 mW cm^−2^ and open circuit voltage of 1.53 V, indicating the Fe-N-CNSs catalysts present a prospective application in practice. In addition, our work provides a novel strategy for further optimizing and improving the catalytic performance of Fe-N-C materials.

## Data Availability

Dare are contained within the article.

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
