# Peer review of "Rational Fabrication of Defect-Rich and Hierarchically Porous Fe-N-C Nanosheets as Highly Efficient Oxygen Reduction Electrocatalysts for Zinc-Air Battery"

_molecules, 2023, doi:10.3390/molecules28072879_

Round 1

Reviewer 1 Report

The paper is well written. However, there are some serious issues as the XRD defects that the complex has not been formed:

1.       In abstract section include your results rather than a theoretical back ground. Also, there are some abbreviations not defined at all. Include concluding remarks at the end of abstract section

2.       Arrange the keyword list alphabetically and omit those words mentioned in manuscript title as they are already reflected in the title.

3.       The units of physical quantities are not properly written throughout the paper. In between the magnitude and unit of a parameter there should be a space with the exception of temp and percent. However, there are spaces in between the units in your manuscript.

4.       There is redundancy of ideas in introduction, present them in a flow and include latest literature. Also include a novelty statement.

5.       In figure 2 a, the single peak presents carbon where are the peaks of Fe. If the catalyst is amorphous then OK otherwise the peak shown present carbon.

6.       What was pyrolysis temp and how nitrogen was retained in the structure after heating at high temperature. What type of proof the author has that it is retained in its structure. It should be confirmed through single crystal XRD and if the interactions are physical then solid state NMR may be required.

Reviewer 2 Report

The writing of this paper is excellent. The manuscript contains some interesting achievements, and the results are well supported with sufficient discussions. However, some minor issues need to be addressed before its acceptance and publication, as listed below:

1.      XRD section, please add the standard hkl bar at the bottom of Figure 2(a).

2.      Is the XRD diagrams coherent with doped content?

3.      Line 224: addition of KSCN, indicating that the C-N sites also contribute to t increased catalytic activity. Remove “t”.

4.      Authors should clearly define all the acronyms and abbreviations when they first appeared, and consistently use them throughout the text.

5.      The manuscript needs a throughout proofreading as there are numerous language and grammar mistakes.

Reviewer 3 Report

The authors reported porous Fe-N-C nanosheets for oxygen reduction reaction (ORR) and zinc-air battery. The Fe-N-CNSs-based zinc-air battery shows significant performance with a peak power density of 128.20 mW cm-2. This work can be considered for acceptance after the following revisions.

1.      The EIS should be performed and fitted and the fitting parameters should be given.

2.      The zinc-air battery performance should be compared with those reported previously in a table.

3.      The open circuit voltage can be mentioned in the abstract and conclusions.

4.      The ORR mechanism on Fe-based catalysts should be discussed. Relevant references in this aspect include but not limits to: Molecules, 2022, 27, 1528; Small Struct., 2023, 2200354; Nano Res., 2022, 15, 38–70; Molecules, 2022, 27, 8644; Chin. J. Catal., 2022, 43, 2057–2090.

Reviewer 4 Report

In this manuscript, Fe-N-CNSs nanosheets with rich defects and hierarchically porous structures were synthesized using metal-organic complexation and mesoporous magnesium carbonate templates. The micro-morphology and fine structure of the catalysts were analyzed by advanced electron microscopy and spectral analysis, and the ORR activity and the discharge performance of zinc-air batteries were evaluated by electrochemical testing. In view of the positive significance of this manuscript for the development of ORR catalysts, I accept this article for publication, provided that the following issues are addressed:

1.     The lattice information of the carbon carrier is calibrated in the HRTEM image. I suggest to select more areas in Fig. 1c for lattice-structure calibration to confirm that graphite carbon exists in (002) crystal plane.

2.     In the manuscript, the author used Raman spectroscopy to analyze the defect degree of the catalyst and proved the positive effect of the defect structure on the ORR activity. The ratio of sp2/sp3 in the XPS C 1s spectrum can also accurately analyze the defect-degree of carbon carrier. Therefore, I suggest adding the analysis of this item.

3.     The positive effects of Fe and C-N species on ORR activity in the catalytic system were verified by KSCN toxicity experiments. The specific types and relative contents of nitrogen-containing species were analyzed by XPS N1s spectroscopy. I suggest to analyze the relationship between the relative content of nitrogen-containing species and ORR activity.

4.        The author mentions " Therefore, designing and exploring cost-effective, efficient, and durable ORR catalysts is vital highly. Metal-nitrogen-carbon (M-N-C) catalysts were already extensively studied as ORR catalysts" in the introduction. M-N-C catalysts exhibit excellent ORR activity due to their unique electronic configuration, and have made important development in recent years. I suggest adding a description of common transition metal based ORR catalysts in the Introduction, and several literatures can be mentioned for reference to enhance the wider readership, e.g., 1) the influence of symmetry of metal states on ORR, https://doi.org/10.3390/sym14122496 ; 2) the optimization of Fe-d band centers, https://doi.org/10.1007/s12274-022-5091-y ; 3) confinement strategy, https://www.science.org/doi/10.1126/sciadv.abn5091.

Round 2

Reviewer 1 Report

ok

Reviewer 4 Report

It can be published now as the authors have made the necessary revisions.